# Prenatal Exposure to Traffic-Related Air Pollution and the DNA Methylation in Cord Blood Cells: MOCEH Study

**DOI:** 10.3390/ijerph19063292

**Published:** 2022-03-10

**Authors:** Jaehyun Park, Woo Jin Kim, Jeeyoung Kim, Chan-Yeong Jeong, Hyesook Park, Yun-Chul Hong, Mina Ha, Yangho Kim, Sungho Won, Eunhee Ha

**Affiliations:** 1Interdisciplinary Program in Bioinformatics, College of Natural Sciences, Seoul National University, Seoul 08826, Korea; spirit9354@snu.ac.kr; 2Department of Internal Medicine and Environmental Health Center, Kangwon National University, Chuncheon 24341, Korea; pulmo2@kangwon.ac.kr (W.J.K.); jeeyoung0628@kangwon.ac.kr (J.K.); jcy6886@naver.com (C.-Y.J.); 3Department of Preventive Medicine, College of Medicine, Ewha Womans University, Seoul 07804, Korea; hpark@ewha.ac.kr; 4Graduate Program in System Health Science and Engineering (BK21 Plus Program), Ewha Womans University, Seoul 07804, Korea; 5Department of Preventive Medicine, College of Medicine, Seoul National University, Seoul 03080, Korea; ychong1@snu.ac.kr; 6Department of Preventive Medicine, College of Medicine, Dankook University, Cheonan 31114, Korea; minaha@dku.edu; 7Department of Occupational and Environmental Medicine, Ulsan University Hospital, University of Ulsan College of Medicine, Ulsan 44033, Korea; yanghokm@ulsan.ac.kr; 8Department of Public Health Sciences, Seoul National University, Seoul 08826, Korea; 9Institute of Health and Environment, Seoul National University, Seoul 08826, Korea; 10RexSoft Inc., Seoul 08826, Korea; 11Department of Occupational and Environmental Medicine, College of Medicine, Ewha Womans University, Seoul 07804, Korea; 12Ewha Medical Research Institute, College of Medicine, Ewha Womans University, Seoul 07804, Korea

**Keywords:** DNA methylation, cord blood, nitrogen dioxide, particulate matter

## Abstract

Particulate matter with a diameter of ≤10 µm (PM_10_) and nitrogen dioxide (NO_2_) affect the DNA methylation in the fetus, but epigenetic studies regarding prenatal exposure to air pollution in Asia are lacking. Therefore, this study aimed to assess whether there is any association between the ambient concentrations of PM_10_ and NO_2_ and CpG methylation in the cord blood DNA by using a Korean birth cohort. The concentrations of the air pollutants were incorporated into the final LUR model by using the maternal address data. The methylation level was determined using HumanMethylationEPIC BeadChip and a linear regression analysis model. A multipollutant model including both PM_10_ and NO_2_ and models with single pollutants were used for each trimester exposure. The number of differentially methylated positions was the largest for midpregnancy exposure in both the single pollutant models and the multipollutant regression analysis. Additionally, gene-set analysis regarding midpregnancy exposure revealed four gene ontology terms (cellular response to staurosporine, positive regulation of cytoskeleton organization, neurotransmitter transport, and execution phase of apoptosis). In conclusion, these findings show an association between prenatal PM_10_ and NO_2_ exposure and DNA methylation in several CpG sites in cord blood cells, especially for midpregnancy exposure.

## 1. Introduction

Traffic-related air pollution, such as particulate matter (PM) and nitrogen dioxide (NO_2_), is one of the major outdoor air pollutions. Exposure to air pollution during pregnancy has pre- and postnatal effects, such as on birth weight and childhood health [1,2].

Ambient air pollution has also been reported to affect DNA methylation [3]. The DNA methylation of cytosine–guanine dinucleotide (CpG) is one of the epigenetic factors that can affect gene expression and is known to be a likely target of environmental exposure [4]. Although maternal smoking is a well-known factor for the DNA methylation of cord blood, the effects of prenatal exposure to traffic-related air pollutants on the genome-wide DNA-methylation statuses of CpG sites have also been reported by birth-cohort studies [5,6]. Furthermore, these effects have been shown to be sex- and trimester-specific [7,8]. In the case of the Asian population, although the associations between ambient concentrations of PM with a diameter ≤ 10 µm (PM_10_) and NO_2_ and DNA methylation have been studied in a Korean adult cohort [9], studies using cord blood are lacking. Although the methylation of several CpGs in this Korean adult cohort was shown to be associated with the PM_10_ and NO_2_ concentrations, studies regarding the effects of prenatal exposure to ambient air pollution on the DNA methylation in infants are lacking.

This study sought to identify epigenomic markers associated with prenatal exposure to ambient air pollutants, particularly NO_2_ and PM_10_, in the cord blood of Korean babies. We used linear regression models to assess the associations between the concentrations of the pollutants and the methylation of each CpG site in the genome of cord blood cells. Additionally, we compared the analyses according to both the trimester during pregnancy and the child’s sex.

## 2. Materials and Methods

### 2.1. Study Population

The Mothers and Children’s Environmental Health (MOCEH) study is a multicenter prospective hospital- and community-based birth-cohort study designed to examine the effects of pre- and postnatal environmental exposures on growth, development, and health from early fetal life to young adulthood in South Korea [10]. From 2006 to 2010, the MOCEH study recruited pregnant women who met the inclusion criteria of being > 18 years of age, <20-weeks pregnant, and a resident of the targeted study site (i.e., Seoul, Ulsan, or Cheonan). The study protocol was approved by the institutional review boards at Ewha Woman’s University (Seoul), Dankook University Hospital (Cheonan), and Ulsan University Hospital (Ulsan, Korea). At the initial prenatal visit, written informed consent was obtained from each participant. Demographic information related to participants’ age, education, income, and socioeconomic status was collected by a structured questionnaire during prenatal visits. Information on infant’s gender, birth weight, gestational age, and birth order was collected using medical records at the time of delivery. Urinary cotinine and creatinine in the pregnant women was measured during their prenatal visits (early and late pregnancy). At birth, cord blood and a piece of tissue from the umbilical cord were obtained. DNA was extracted from the blood samples into tubes containing ethylenediaminetetraacetic acid. Among the 1751 base participants recruited before week 20 of pregnancy, 383 cord blood samples corresponding to 195 baby boys and 188 baby girls were selected to undergo DNA methylation profiling and were finally included in the present study.

### 2.2. Assessment of Air-Pollutant Concentration

The exposure of each participant to the air pollutants was assessed on the basis of their residential address by using geographical information system (GIS) variables and estimated NO_2_ and PM_10_ concentrations from a land-use regression (LUR) model, which is a standardized method described previously [11]. We first obtained the monthly concentrations of PM_10_ and NO_2_ measured at all national atmospheric monitoring stations throughout Korea (https://www.airkorea.or.kr/eng) (accessed on 27 Nov 2019) from 2006 to 2011. The number of national air quality monitoring stations in 2006 was 202 and reached 246 by 2011. The residential addresses of the study subjects were converted into Transverse Mercator (TM) coordinates and mapped. This map was superimposed with the national road-network and land-use maps (1:25,000 scale). Using GIS, the lengths of all the roads within a 200 m buffer zone, the traffic intensity at the nearest road, the total heavy-duty traffic loads on all the roads within a 100 m buffer zone, the green area within 300 m of the individual’s residence, and the altitude of the residence were then calculated. Finally, individual exposures were estimated by incorporating these values into the final LUR model, which included the lengths of all the roads, traffic intensity at the nearest road, total heavy-duty traffic loads on all the roads, and variables representing the spatial trends derived from GIS (ArcGIS version 9.3, ESRI, Redlands, CA, USA). We then calculated the monthly exposure levels at the addresses of the participants in the three regions (Seoul, Ulsan, and Cheonan) by using a modeling method with a GIS. LUR analyzes traffic-related air-pollution exposure values through a multiple linear regression modeling method between predictors that can best explain traffic-related air-pollution concentration among variables such as total road length and traffic intensity on roads [12]. In the modeling process, we analyzed how the air-pollution concentration value was predicted through the LOOCV method and RMSE confirmation. Multiple linear regression models were built using a supervised forward stepwise procedure. Leave-one-out cross-validation (LOOCV) was used to evaluate the model performance, and the overall fit (R^2^) and root mean squared error (RMSE) between the predicted and measured concentrations per site were calculated. The models used the centrally and locally available geographic variables as potential predictors. The following five predictor variables were left in the final LUR model for NO_2_: the total length of all the roads within 300 m, traffic intensity on the nearest road, total heavy-duty traffic load of all the roads within 100 m, urban green area within 300 m, and a variable representing the spatial trend. The R^2^ of the model and the LOOCV R^2^ of the NO_2_ models were 0.79 and 0.73, respectively, indicating that the LUR model explained a large fraction of the measured NO_2_ spatial variability based on the monitoring data. The predictor variables that best described the spatial variation in the PM_10_ concentration (R^2^ = 0.69 and LOOCV R^2^ = 0.60) included the total length of all the roads within 300 m, total heavy-duty traffic load within 100 m, and the spatial trend variable.

The trimester-specific mean concentrations of PM_10_ and NO_2_ were calculated as the mean pollutant concentration associated with the length of each pregnancy per trimester. Following this procedure, we estimated the exposure levels during three periods of pregnancy (0–12 weeks (first trimester), 12–27 weeks (second trimester), and 27–40 weeks (third trimester)) and during the period from conception until delivery (entire pregnancy).

### 2.3. Data Extraction, Cleaning, and Imputation

The missing values of adjusting covariates were imputed with the missForest package in R [13]. A pair of samples were twins of the same sex and maternal variables, and thus they were excluded from the imputation. Therefore, 383 nonredundant samples with detailed information on the corresponding infant sex, smoking history survey, and maternal blood concentrations of cotinine and creatinine in the early and later pregnancy were included in the imputation step. Subjects with high concentrations (≥200 µg/gm) of maternal urinary cotinine in the later pregnancy or with a positive response to the question on current smoking were grouped as “current smokers”.

Preprocessing of the information on the cord blood samples, leukocyte-composition estimation, and quality-control steps were performed with the R packages ewastools [14,15] and minfi [16], as described previously [17]. Briefly, leukocyte compositions were estimated with Houseman’s method, with the reference panel by Salas et al. with IDOL-optimized CpGs [18]. Quality-control steps were performed by (1) BeadArray Controls metrics by Illumina, (2) sex discordance by the probe intensities, (3) outliers or duplicates by SNP probes, (4) outliers by principal components (PCs), and (5) outliers by leukocyte compositions. The quality-control results are shown in the Appendix A.

### 2.4. Genotyping and Assessment of the Methylation Levels

The methylation levels of the cord blood samples were determined using Illumina HumanMethylationEPIC BeadChip with the manufacturer’s instructions. Genotyping of the samples was performed as previously described [19]. Briefly, cord blood samples were genotyped on Asian Precision Medicine Research Array (APMRA; Affymetrix, CA, USA). Quality control was conducted using plink v1.90b3.44 [20] and WISARD 1.3.3, with the criteria of X chromosome inbreeding coefficients, missing rates, identity-by-state (IBS) matrix, multidimensional scaling (MDS) plots, Hardy–Weinberg equilibrium (HWE), and minor allele frequencies. The rest of the missing values were filled with SHAPEIT v2.r837 [21], and the data were then imputed to the whole-genome scale by using IMPUTE 2.3.2 with the phase-3 reference data from the 1000 Genome Project [22,23,24]. The same quality-control procedures were conducted once again after the imputation step. The results of the quality-control step are described in the Appendix A.

### 2.5. Epigenome-Wide Association Analysis (EWAS)

We used Linear Models for Microarray Data (Limma) to perform the linear regression analysis, with the beta values as dependent variables. The CpG sites with >3% of the missing rates in any of the acquisition times were excluded from the analyses. Infant sex, maternal current smoking status, and the estimated leukocyte composition were used as adjusting covariates, and the batches were adjusted using the random effect variable. Because some of the data were positive and right-skewed, the air-pollutant variables were log_10_-transformed before the analyses.

As the main analysis, regression models with both of the air pollutants were used, and these models were defined as “multipollutant models”. As a complement, analyses with models including one of the pollutants were also performed. The above procedures were conducted with the air-pollution concentrations during (1) the whole pregnancy, (2) the first trimester of pregnancy, (3) the second trimester of pregnancy, and (4) the third trimester of pregnancy.

### 2.6. Reverse Causation Identification

To identify the false positives of the differentially methylated positions (DMPs), we ran instrumental-variable regression analyses using the R package AER [25]. The genotype data were used as instrumental variables (IVs). To avoid the weak-instrument problem, we first ran linear regression analyses with the pollution as the response variable by using plink v1.90b3.44 [20], where the infant sex, maternal current smoking statuses, estimated leukocyte compositions, and 10 genotype principal component (PC) scores were included as adjusting covariates. We used the results with *p*-values < 0.05.

In the IV regression, methylation and pollution variables were set as treatment and outcome variables, respectively. The consistency of the estimators from the regression was tested with Wu–Hausmann test. The results with Wu–Hausman test *p*-values < 0.05 were ignored since the estimators from the IV regressions were not consistent and the test may not have been informative. The alternative hypothesis of the IV regression was that the treatment had a causal effect on the outcome. In other words, the significant results with regression *p*-values < 0.05 indicated that there were significant causal effects on air pollution due to the DNA methylation statuses. Since this does not make sense, the corresponding results were considered as false positives and excluded from the interpretation.

## 3. Results

### 3.1. General Characteristics and Exposure to Ambient Air Pollutants

Three hundred and fifty-eight samples with complete profiles of air-pollutant exposure and methylation statuses were used in the analysis. Among them, 180 were male infants and 178 were female infants, and 7 mothers (1.96%) were current smokers. The geometric means of exposure to NO_2_ and PM_10_ during the entire pregnancy were 0.0260 ppm (10th percentile = 0.0186; 90th percentile = 0.0417) and 53.49 μg/m^3^ (10th percentile = 45.74; 90th percentile = 61.58), respectively (Table 1). The summary statistics of the air-pollutant variables, both overall and trimester-specific, and the adjusting covariates are shown in Table 2.

We found that some of the pollutant variables were right-skewed (Figure 1). Since the performance of linear regression analysis depends on the normality of variables, we tried to alleviate the skewness and make the variables closer to the normal distribution by applying log_10_ transformation. We calculated the skewness and kurtosis of the variables, which were 0 and 3, respectively, if the variables were normally distributed, before and after the transformation. For NO_2_ variables, we found that the skewness became closer to 0 and less skewed after the transformation (Table 3). Although some of the variables did not show a meaningful difference or became worse after the transformation, we comprehensively applied the log_10_ transformation to all of the variables for the consistency of the analysis.

### 3.2. Multipollutant Analysis

The correlations between NO_2_ and PM_10_ measured by Pearson’s and Spearman’s correlation coefficients in each period were around 0.5, and the variance inflation factors for these variables from the multipollutant linear models ranged from 1.40 to 1.51. Therefore, we judged that the multicollinearity would not interrupt the performance of the analyses. As many of the analyses showed deflated *p*-values, we adjusted the *p*-values via Bonferroni’s method, which is the most conservative method.

#### 3.2.1. Whole Pregnancy

We found two CpG sites in chromosome 10 showing significant relationships with the air-pollution exposure during the whole pregnancy (Table 2). One of them, cg14547404 (*p*-value, 1.78 × 10^−11^), showed a significant correlation with the concentration of PM_10_ (*p*-value, 4.31 × 10^−11^) but also showed significant results from the IV regression (*p*-value, 7.52 × 10^−3^; Wu–Hausman test *p*-value 0.330), and is thus presumably a false signal. The other, cg06517429 (*p*-value, 2.37 × 10^−8^), was in the 5′ UTR or on the 1st exon of *CASP7*, depending on the isoforms, and in the CpG island at position 115,439,007-115,440,196 and showed a significant correlation with NO_2_ (*p*-value, 5.05 × 10^−8^). There was also a significant association between NO_2_ and cg08096307 (*p*-value, 5.81 × 10^−8^), a site in a CpG island at position 221,064,889-221,065,600 of chromosome 1, and this association was also significant (*p*-value, 1.81 × 10^−7^) when both of the pollutants were simultaneously tested.

Further, when we conducted the analyses with only the female infants, cg14547404 and cg00670246 were found to be significantly correlated with the pollutants (*p*-values, 4.36 × 10^−10^ and 4.44 × 10^−10^, respectively). The cg00670246 was in gene *GRHL1*. Both of these sites showed significant results with PM_10_ (*p*-values, 4.93 × 10^−9^ and 1.49 × 10^−9^, respectively). Additionally, IV regression from cg14547404 to PM_10_ showed significant results (*p*-value, 0.0251; Wu–Hausman test *p*-value, 0.758). PM_10_ was also correlated with cg14561322 and cg02737288 (*p*-values, 1.47 × 10^−7^ and 2.75 × 10^−7^, respectively) and also showed low *p*-values when both of the pollutants were tested.

There was no significant correlation between the pollutants and male infants. However, testing for only the regression coefficient of NO_2_ revealed one significant site, cg19390934 (*p*-value, 3.86 × 10^−8^), which was of genes *MTERFD2* and *SNED1* (Table 4).

#### 3.2.2. The First Trimester of Pregnancy

The exposure to air pollutants in early pregnancy did not show any significant correlation with the methylation statuses. However, testing for only the coefficient of NO_2_ showed notable results with cg19190403 (*p*-value 1.23 × 10^−8^) and cg06517429 (*p*-value 2.65 × 10^−8^). The site cg19190403 did not have any annotation. Analyzing with only the male samples revealed that cg27535616 was significantly correlated with the NO_2_ concentration in early pregnancy (*p*-value 3.99 × 10^−8^; Table 5).

#### 3.2.3. The Second Trimester of Pregnancy

Air-pollutant exposure in midterm pregnancy showed the largest number of DMPs (Table 6). We observed more significant correlations with NO_2_ than PM_10_ concentrations in individual analyses. The site cg06517429 (*p*-value 3.47 × 10^−8^), one of the main findings from the analyses of whole pregnancy, was also found to be significantly correlated to air pollutants in the second trimester. The most significant sites included cg04129282 (*p*-value 4.37 × 10^−9^) on chromosome 15, annotated with *WDR93* and *PEX11A*, and cg06772824 (*p*-value 4.84 × 10^−9^) on chromosome 2, annotated with *FAM176A*.

When the analyses were conducted separately according to infant sex, one CpG site (cg14262371) was significantly associated with the air pollutants in the female infants (*p*-value 3.54 × 10^−8^). The site cg14262371 is in the 5ʹ UTR or the 1st exon of *MOV10* and the N-shore of a CpG island at position 113,217,475-113,218,097 of chromosome 1.

#### 3.2.4. The Third Trimester of Pregnancy

The site cg06517429 was significantly correlated with the concentration of NO_2_ (*p*-value 5.46 × 10^−8^), again in the analysis in the later pregnancy. In the female infants, a new site named cg20654468 was found to be related to the concentration of PM_10_ (*p*-value 4.58 × 10^−8^). This site was annotated to be in *LPXN* and the N-shelf of a CpG island at position 58,345,673-58,347,321 of chromosome 11 (Table 7).

### 3.3. Models with a Single Pollutant

Results of LIMMA analyses with the models including only NO_2_ or PM_10_ are tabulated in Appendix A. We found that cg06517429 from the multipollutant models was identified in the NO_2_ whole-pregnancy model, while cg14547404 from the multipollutant model was identified in the PM_10_ single-pollutant model. Additionally, similar to the results from the multipollutant models, air-pollutant exposure in the second trimester of pregnancy had the largest effect on the methylation statuses of the infants.

### 3.4. Sensitivity Analysis

We added family income in the EWAS analyses as a covariate to check whether socioeconomic condition plays major role in the epigenetic alterations by air-pollution exposure. After including family income, while cg14547404 was still significant (1.78 × 10^−11^), cg06517429 and cg08906307 became borderline for the whole pregnancy (2.38 × 10^−7^ and 1.24 × 10^−6^, respectively, Appendix A).

### 3.5. MRC-IEU EWAS Catalog

Several phenotypes have previously been reported to be related to the methylation statuses of the DMPs we identified, according to the MRC-IEU EWAS catalog, and the data are listed in the Appendix A. Most of the data were from the ARIES consortium, and diverse phenotypes, such as omega 6 fatty acid ratio (cg06517429) and hip cortical ratio (cg06772824), were reported. Moreover, Gross et al. detected several CpG sites, such as cg19390934, cg16274061, cg00894435, and cg20654468, to be differentially methylated by chronic HIV infection. In addition, several studies have reported significant correlations between our DMPs and some clinical traits, such as cg19390934 and smoking status, cg04129282 and rheumatoid arthritis, and cg20654468 and smoking status or child age.

## 4. Discussion

In this study, we identified several CpG sites that were associated with prenatal exposure to PM_10_ and NO_2_. We found that the largest number of CpGs were associated with the pollutants exposed to in the midterm pregnancy. We also observed sex-specific associations with the pollutants.

The CpG island near the *ARHGAP22* gene (cg14547404) was associated with average PM_10_ concentration during the whole pregnancy period. This gene was also the most significant DMP in the female subgroup. *ARHGAP22* is known to control cell morphology by regulating the actin cytoskeleton [26].

The CpG islander *CASP7* gene (cg06517429) was significantly associated with average NO_2_ concentration during all the trimesters. This gene is related to apoptosis [27]. The methylation of this CpG site is associated with *CASP7* expression in the adipose tissue [28]. *CASP7* is a critical mediator of mitochondrium-induced apoptosis [29], and interestingly, a previous study revealed that NO_2_ exposure during pregnancy is associated with differences in the cord blood methylation patterns of several genes involved in mitochondrial function [5]. Furthermore, these in utero influences seem to persist into early childhood. The effects of NO_2_ concentration on the CpGs near the *CASP7* gene may also need to be analyzed after birth.

The vulnerable period may be different between girls and boys [30]. In the present study, a few sites were significantly detected in association with exposure during whole and late pregnancy in only female infants.

There is a lack of consensus regarding the most sensitive time during the prenatal period for air-pollution exposure [31,32,33]. In the current study, more CpG sites were associated with pollutant exposure during the second trimester than the other periods. A recently published study found that PM_10_ exposure during the second trimester is associated with decreased head circumference, and low birth weight and small size for gestational age are associated with PM_10_ exposure in the third trimester [32]. Therefore, the correlation between the CpG sites found in the present study and their functional contribution to a specific period should be analyzed. Compared with the PM_10_ levels, the NO_2_ levels were correlated with more CpG sites in the results of the trimester-specific analysis. We reported results using multipollutant models as the main results. The single pollutant model also showed similar results that support the largest effect of the pollutants during the second trimester on the methylation statuses of the infants.

We found that the proportion of natural killer cells in our samples were relatively low (3.31%; Table 2). This may be due to the selection of the leukocyte deconvolution methods, and it may differ by the method or the reference panel. Unfortunately, we could not test other deconvolution methods and compare the results due to the shortage of time. Further studies may investigate the effect of the methods on the overall results. Furthermore, placental DNA methylation is another organ reflecting prenatal environmental exposure and potentially related to health outcomes, which can be compared to cord blood results [34].

There are a few limitations of this study. The results need to be replicated in other studies. Several CpGs associated with air pollution in this study have also been reported to be associated with chronic HIV infection or smoking [35]. When we attempted to search the PACE data of NO_2_ and PM exposure [5,6], no replicated result was detected. However, we identified that similar pathways were enriched with genes near the significantly associated CpG sites. Second, we did not quantitate gene expression. Further transcription information may facilitate finding meaningful pathways relevant to the sites influenced by air pollutants [36]. Third, air-pollution levels at residential addresses might not represent personal exposure levels. Some individuals may prefer spending much of their daily time at other places, such as the workplace, instead of the residential addresses. However, misclassification in the degree of personal exposure level can be assumed to be nondifferential according to the methylation levels, which cannot alter the observed association but can weaken it. Our results should also be interpreted while keeping in mind the obscured variances that were unmeasured.

In conclusion, prenatal exposure to PM_10_ and NO_2_ is associated with several CpG sites on the genomic DNA of cord blood cells. Therefore, exposure to traffic-related air pollutants, such as PM_10_ and NO_2_, during pregnancy, especially in the midtrimester, can affect biological events by altering the methylation statuses of certain CpG sites.

## Figures and Tables

**Figure 1 ijerph-19-03292-f001:**
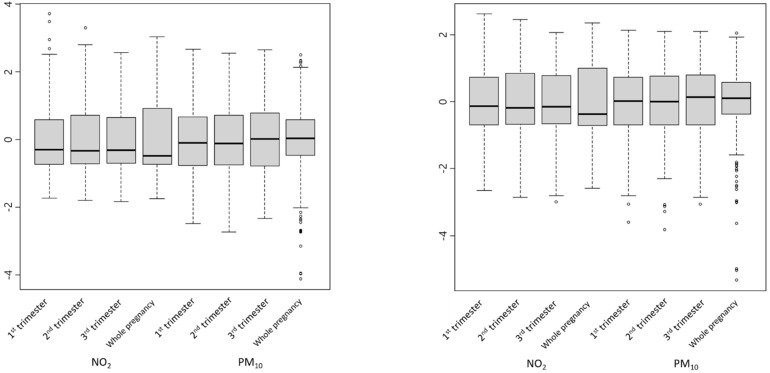
Boxplots of pollution variables before (**left**) and after (**right**) log10 transformation, drawn after centering and scaling.

**Table 1 ijerph-19-03292-t001:** The geometric means of air-pollution-exposure variables in the 358 subjects.

Air Pollutants (Unit)	Pregnancy			
	Whole Period Mean (min–max)	1st Trimester Mean (min–max)	2nd Trimester Mean (min–max)	3rd Trimester Mean (min–max)
NO_2_ (ppm)	0.0260 (0.0113–0.0559)	0.0249 (0.0098–0.0628)	0.0261 (0.0098–0.0609)	0.0264 (0.0091–0.0557)
PM_10_ (ug/m^3^)	53.49 (24.96–71.93)	52.30 (21.47–86.38)	52.02 (23.09–81.23)	53.71 (26.03–83.06)

The geometric means of air-pollution-exposure variables in the 358 subjects. The minimum/maximum values are also listed in the parentheses. The concentrations of the pollutants were estimated using the LUR model incorporated with the residential address of each subject.

**Table 2 ijerph-19-03292-t002:** Summary statistics of the adjusting covariates in the 358 subjects. For clinical covariates, the number and the proportion are shown. For leukocyte compositions, the average values are shown, and the values in the parentheses indicate the standard errors.

Variables	Summary
**Covariates, N (%)**	
Child Sex	
Male	180 (50.28%)
Female	178 (49.72%)
Maternal smoking, current smokers	7 (1.96%)
**Estimated leukocyte composition of** **the cord blood samples**	
CD4 T cells (%)	20.70 (0.449)
CD8 T cells (%)	4.46 (0.173)
Natural killer cells (%)	3.31 (0.141)
B cells (%)	5.18 (0.130)
Granulocytes (%)	55.86 (0.651)
Monocytes (%)	9.34 (0.184)
Nucleated red blood cells (%)	3.08 (0.241)

**Table 3 ijerph-19-03292-t003:** Skewness and kurtosis of pollution variables before and after log10 transformation.

	NO_2_				PM_10_			
	Whole Pregnancy	1st Trimester of Pregnancy	2nd Trimester of Pregnancy	3rd Trimester of Pregnancy	Whole Pregnancy	1st Trimester of Pregnancy	2nd Trimester of Pregnancy	3rd Trimester of Pregnancy
**Before transformation**								
Skewness	0.756	0.944	0.869	0.828	−0.682	0.331	0.221	0.021
Kurtosis	2.386	3.478	2.864	2.640	4.949	2.477	2.660	2.194
**After transformation**								
Skewness	0.315	0.161	0.280	0.210	−1.585	−0.230	−0.357	−0.420
Kurtosis	2.195	2.624	2.361	2.470	8.461	2.800	3.232	2.498

**Table 4 ijerph-19-03292-t004:** Differentially methylated positions (DMPs) in cord blood DNA in association with exposure to multiple ambient pollutants during whole pregnancy.

CpG ID	*p*-Value (All Pollutants)	Log-FC (NO_2_)	*p*-Value (NO_2_)	Log-FC (PM_10_)	*p*-Value (PM_10_)	Gene Annotation	CpG Island Annotation
All infants (*N* = 358)							
cg14547404	**1.78 × 10^−11^**	−0.0177	0.237	0.231		*ARHGAP22* TSS1500/body	chr10: 49863620–49864601 N-shore
cg06517429	**2.37 × 10^−8^**	0.0664	**5.05 × 10^−8^**	−0.0195	0.472	*CASP7* 5′ UTR/1st exon	chr10: 115439007–115440196 island
cg08906307	1.81 × 10^−7^	0.0641	**5.81 × 10^−8^**	−0.0440	0.0959	-	chr1: 221064889–221065600 island
Male infants (*N* = 180)							
cg19390934	2.60 × 10^−7^	−0.0577	**3.86 × 10^−8^**	0.0699	0.0124	*MTERFD2* body *SNED1* body	-
Female infants (*N* = 178)							
cg14547404	4.36 × 10^−10^	−0.0112	0.619			*ARHGAP22* TSS1500/body	chr10: 49863620–49864601 N-shore
cg00670246	4.44 × 10^−10^	−0.0315	0.309	0.405		*GRHL1* body	-
cg14561322	1.47 × 10^−7^	−0.0482	1.81 × 10^−3^	0.184		-	-
cg02737288	2.75 × 10^−7^	−0.0504	3.90 × 10^−3^			*SGPP1* body	Unknown

*p*-values for “all pollutants” indicate the *p*-values from F-tests for both of the pollutants, and *p*-values for each pollutant mean the *p*-values of the corresponding regression coefficients from the multipollutant model. *p*-values that reached the significance cutoff (Bonferroni adjusted *p*-value < 0.05) are marked in bold font. The significant results from reverse causation analyses (instrument variable regression *p*-value < 0.05 and Wu–Hausman test *p*-value ≥ 0.05) are considered as false positives and marked with strikethroughs. TSS1500, ~1.5 kb upstream of the transcription start site (TSS); UTR, untranslated region; N-shore, ~2 kb upstream of the CpG island.

**Table 5 ijerph-19-03292-t005:** Differentially methylated positions (DMPs) in cord blood DNA in association with exposure to multiple ambient pollutants during the first trimester of pregnancy.

CpG ID	*p*-Value (All Pollutants)	Log-FC (NO_2_)	*p*-Value (NO_2_)	Log-FC (PM_10_)	*p*-Value (PM_10_)	Gene Annotation	CpG Island Annotation
All infants (*N* = 358)							
cg19190403	8.38 × 10^−8^	−0.0478	**1.23 × 10^−8^**	0.0354	4.76 × 10^−3^	-	-
cg06517429	9.37 × 10^−8^	0.0641	**2.65 × 10^−8^**	−0.0343	0.0456	*CASP7* 5′ UTR/1st exon	chr10: 115439007–115440196 island
Male infants (*N* = 180)							
cg27535616	2.59 × 10^−7^	0.0974	**3.99 × 10^−8^**	−0.0660	0.0123	Unknown	Unknown

*p*-values for “all pollutants” indicate the *p*-values from F-tests for both of the pollutants, and *p*-values for each pollutant mean the *p*-values of the corresponding regression coefficients from the multipollutant model. *p*-values that reached the significance cutoff (Bonferroni adjusted *p*-value < 0.05) are marked in bold font. UTR, untranslated region.

**Table 6 ijerph-19-03292-t006:** Differentially methylated positions (DMPs) in cord blood DNA in association with exposure to multiple ambient pollutants during the second trimester of pregnancy.

CpG ID	*p*-Value (All Pollutants)	Log-FC (NO_2_)	*p*-Value (NO_2_)	Log-FC (PM_10_)	*p*-Value (PM_10_)	Gene Annotation	CpG Island Annotation
All infants (*N* = 358)							
cg04129282	**4.37 × 10^−9^**	0.0217	9.61 × 10^−4^	0.0334	2.12 × 10^−3^	*WDR93* TSS1500 *PEX11A* body	chr15: 90233530–90234083 island
cg06772824	**4.84 × 10^−9^**	0.0278	4.29 × 10^−6^	0.0144	0.146	*FAM176A* TSS200/5′ UTR	chr2: 75787717–75788312 island
cg03233931	**2.85 × 10^−8^**			8.09 × 10^−3^	0.166	*LRRC20* TSS200/5′ UTR/ body	chr10: 72141560–72142637 island
cg06517429	**3.47 × 10^−8^**	0.0622	6.83 × 10^−8^	−0.0119	0.523	*CASP7* 5′ UTR/1st exon	chr10: 115439007–115440196 island
cg16274061	**4.50 × 10^−8^**	0.0252	1.30 × 10^−3^	0.0355	6.14 × 10^−3^	*SAP30L* TSS1500	chr5: 153825417–153826526 N-shore
cg23560755	**5.06 × 10^−8^**			0.0135	0.123	*SORBS3* body	chr8: 22422534–22423702 N-shelf
cg00894435	**5.60 × 10^−8^**	0.0337	4.76 × 10^−4^	0.0380	0.0169	*SV2B* TSS1500	chr15: 91642908–91643702 island
Female infants (*N* = 178)							
cg14262371	**3.54 × 10^−8^**	0.0150	2.97 × 10^−4^			*MOV10* 5ʹ UTR/1st exon	chr1: 113217475–113218097 N-shore

*p*-values for “all pollutants” indicate the *p*-values from F-tests for both of the pollutants, and *p*-values for each pollutant mean the *p*-values of the corresponding regression coefficients from the multipollutant model. *p*-values that reached the significance cutoff (Bonferroni adjusted *p*-value < 0.05) are marked in bold font. The significant results from reverse causation analyses (instrument variable regression *p*-value < 0.05 and Wu–Hausman test *p*-value ≥ 0.05) are considered as false positives and marked with strikethroughs. TSS1500, ~1.5 kb upstream of the transcription start site (TSS); TSS200, ~200 kb upstream of the TSS; UTR, untranslated region; N-shelf, ~4 kb upstream of the CpG island; N-shore, ~2 kb upstream of the CpG island.

**Table 7 ijerph-19-03292-t007:** Differentially methylated positions (DMPs) in cord blood DNA in association with exposure to multiple ambient pollutants during the third trimester of pregnancy.

CpG ID	*p*-Value (All Pollutants)	Log-FC (NO_2_)	*p*-Value (NO_2_)	Log-FC (PM_10_)	*p*-Value (PM_10_)	Gene Annotation	CpG Island Annotation
All infants (*N* = 358)							
cg06517429	2.92 × 10^−7^	0.0616	**5.46 × 10^−8^**	−0.0399	0.0206	*CASP7* 5ʹ UTR/1st exon	chr10: 115439007–115440196 island
Female infants (*N* = 178)							
cg20654468	2.80 × 10^−7^	−0.0398	6.87 × 10^−3^	0.123	**4.58 × 10^−8^**	*LPXN* body	chr11: 58345673–58347321 N-shelf

*p*-values for “all pollutants” indicate the *p*-values from F-tests for both of the pollutants, and *p*-values for each pollutant mean the *p*-values of the corresponding regression coefficients from the multipollutant model. *p*-values that reached the significance cutoff (Bonferroni adjusted *p*-value < 0.05) are marked in bold font. UTR, untranslated region; N-shelf, ~4 kb upstream of the CpG island.

## Data Availability

The data presented in this study are available on request from the corresponding author. The data are not publicly available due to privacy issues.

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
