# Peer review of "Prenatal Exposure to Traffic-Related Air Pollution and the DNA Methylation in Cord Blood Cells: MOCEH Study"

_ijerph, 2022, doi:10.3390/ijerph19063292_

Round 1
Reviewer 1 Report
See attached file for comments.

Author Response
It is now widely recognized that there is another class of “environmental” factors capable of playing a relevant role in epigenetic alterations and adverse birth outcomes. In order to test the association between air pollution and molecular endpoints (i.e. DNA methylation of cord blood cells), it is also essential to take into account the epigenetic influence of socioeconomic condition on the subjects exposed to air pollution. The manuscript ignores this aspect which would have deserved to be included in the study design. Authors should fill this gap (albeit only partially) by providing a compelling description of the reasons why the socioeconomic condition of the subjects enrolled in the study was not explored, given the role it could play as a potential confounder or modifier of the association between exposure to atmospheric pollutants and epigenetic alterations.
Response: According to reviewer’s comment,
- we have added the information in the study design section 2.1 (line 89-93)
- we performed sensitivity analysis after including socioeconomic status and our results were similar. We added results in supplementary material 2, and mentioned in the Results section. (Line 299-303)
“Demographic information related to participants’ age, education, income and socio-economic status was collected by a structured questionnaire during prenatal visits. Information on infant’s gender, birth weight, gestational age and birth order was collected using medical records at the time of delivery. Urinary cotinine and creatinine in the pregnant women was measured during their prenatal visits (early and late pregnancy). ”
“We added family income in the EWAS analyses as a covariate to check whether socioeconomic condition plays major role in the epigenetic alterations by air pollution exposure. After including family income, while cg14547404 was still significant (1.78x10-11), cg06517429 and cg08906307 became borderline for the whole pregnancy (2.38x10-7 and 1.24x10-6, respectively, Supplementary material 2)”
I wonder if the English text was revised, possibly by a professional, as I found some sentences that sound not formally correct, such as the one at lines 75-76.
Response: We corrected the English text at lines 74-75 as follows.
“Additionally, we complemented analyses by both trimester during pregnancies and child’s sex.”.
All the citations within the text must be arranged in their layout, particularly inserting a blank before the left parenthesis and separating the name of the first author from the word “et”. For example, line 56 should be “health (Cai et al. 2020; Lee et al. 2020)”, and so on.
Response: We corrected all the citations.
Specific points:
Abstract
Line 35: “with a diameter of < 10 um” should be “with a diameter ≤ 10 um”.
- Introduction
Line 65: also here, “with a diameter of < 10 um” should be “with a diameter ≤ 10 um”.
Response: We have corrected the symbols in abstract and introduction section.
- Materials and Methods
Lines 90-92: please explain why only 383 cord blood samples among the 1,751 base participants recruited were included in this study.
Response: We performed DNA methylation profiling in another project, thus we had to select limited number of cord blood samples from the main cohort. We explained in the methods section (Line 96-98) as follows:
“383 cord blood samples corresponding to 195 baby boys and 188 baby girls were selected to be undergone DNA methylation profiling and finally included in the present study.”
Line 99 and 188: the web sites reported in parenthesis should be uniform with the text.
Response: We removed the underline in parenthesis.
Line 133-134: a reference is required for missForest package in R; for example, “Daniel J. Stekhoven (2013) missForest: Nonparametric Missing Vale Imputation using
Response: We added reference for missForest package in R.
Line 142: again, a reference is required for R packages ewastools and minfi
Response: We added references for ewastools and minfi package in R.
Line 146: the citation “Park et al (Park et al.2020)” should be rearranged
Response: We rearranged the citation.
Line 159-161: Infant sex, maternal current smoking status, and the estimated leukocyte composition were used as adjusting covariates….”; in the Introduction, the reason why these variables affect the end-point should be justified.
Response: We added text in the introduction (Line 60-61) as follows: “Although maternal smoking is well-known factor for DNA methylation of cord-blood DNA methylation”
Line 162-163: please indicate if after log-transformation the distribution became normal.
Response: We have added boxplots and tables of skewness and kurtosis of pollutant variables before and after log-transformation. Also, we explained the reason of the transformation in the results section (3.1 line 226-234).
“We found that some of the pollutant variables are right-skewed (Figure 1). Since the performance of linear regression analysis depends on the normality of variables, we tried to alleviate the skewness and to make variables closer to normal distribution by applying log10-transformation. We calculated the skewness and kurtosis of the variables, which are 0 and 3 respectively, as the variables are normally distributed, before and after the transformation. For NO2 variables, we found that the skewness became closer to 0 and less skewed after the transformation (Table 3). Although some of the variables did not show meaningful difference or became worse after the transformation, we comprehensively applied the log10-transformation to all of the variables for the consistency of the analysis.”
Line 165: please supply a reference for “multi-pollutant models”.
Response: We apologize for the confusing text. We meant to define these models as “multi-pollutant models” and have changed the text (Line 186).
Line 180-181: the sentence should be rewritten more clearly and correctly, specifying what the null hypothesis was; furthermore, to state that the null hypothesis is accepted; nor it is correct to define the result as "significant". Also for Wu-Hausman test the statistical significance is obtained when the p-value is < 0.05. If, on the other hand, this significance is not reached because the p-value is ≥ 0.05, the authors can infer what they deem most appropriate on the endogeneity of a variable, but without distorting the meaning of the p-value itself, attributing significance where it does not exist.
Response: We have rewritten the sentences to make the meaning of the tests clearer (line 201-208), as follow:
“The consistency of the estimators from the regression was tested with Wu-Hausmann test. The results with the Wu-Hausman test p-values < 0.05 were ignored since the estimators from the IV regressions are not consistent and the test may not be informative. The alternative hypothesis of the IV regression is that the treatment has causal effect on the outcome. In other words, the significant results with the regression p-values < 0.05 indicate that there were significant causal effects on air pollution due to the DNA methylation statuses. Since this does not make sense, the corresponding results were considered as false positives and excluded from the interpretation.”
Line 189-190: again, a reference is required for R package missMethyl.
Response: We added reference for missMethyl package in R.
Line 192-194: “Only the GO terms in the biological process (BP) category with ≥ 5 genes and the KEGG pathways with ≥ 5 genes were tested. The GSA was performed only if there were ≥ 1 DMP”; please justify the used of these criteria, possibly with supporting bibliography.
Response: We agree that the enrichment tests can be instable, if the number of genes in the term is too low. We removed GSA analysis from the manuscript as reviewer 3 recommended.
- Results
Line 203-204: why “Table 1” is written in bold? The same for the other tables cited in the text (lines 236, 252, 261, 282, 290, 299, 315).
Response: We converted the bold text to the plain text for all the cited tables.
Line 212: the acronym for VIF is not defined.
Response: We wrote the full form of VIF.
Line 257-258: maybe these two sentences are pertaining to the notes of Table 3, so they should be linked with the preceding text, also changing the format of the character to avoid any confusion with the main text.
Response: We modified the text and provided results from multi-pollutant as the preceding text.
“We found that cg06517429 from the multi-pollutant models was identified in NO2 whole pregnancy model, while cg14547404 from the multi-pollutant model was identified in the PM10 single-pollutant model.”
Discussion
Why this section is not numbered as the preceding ones?
Response: We provided number 4 for the discussion section and did not make subtitles.
Line 332: there is a point to remove after “The CpG island”.
Response: We removed the point.
Line 370: PACE data should be defined
Response: We added references for the PACE studies. (Grieve et al. 2017; Gruzieva et al. 2019)
Table
Please check and rearrange the layout of the tables, to make them easy to read and more comprehensible; in particular Table 1 (where some special symbols appear), Tables 2 and 4 (where especially last column is confusing).
Response: We separated the Table 1 into 2 tables and moved the footnotes to the table descriptions for readability. Also, for the tables corresponding to the results, we arranged the sentences in footnotes and moved them to the table descriptions.
Table 1: it would be better to use the terms “min” and “max” instead of :range” (which it is expected to be a single number, calculated as the difference between maximum and minimum value); moreover, the values of geometric means and of minimum and maximum for NO2 and for PM10 during the whole pregnancy seem not coherent with the same data reported for the three trimester separately; please check and amend where necessary.
Response: We changed the term “range” to “minimum/maximum values”. Minimum and maximum NO2 and PM10 were assessed during entire pregnancy.
The range for each trimester and the whole period can be different. For example, in the case of the subject with the lowest NO2 concentration in the 1st trimester (0.0098), the NO2 concentration in the 2nd and 3rd trimester was 0.0156 and 0.0191, respectively, and the concentration in whole pregnancy was 0.0149 (the difference from the actual arithmetic/geometric mean appear to be due to the result of LUR regression).
Further explanation, the reason why whole pregnancy range may be different from the range by trimester is that the whole pregnancy concentration is close to the average value of the concentration by trimester. This is also presented by the fact that the whole pregnancy range is narrower than the range by trimester.
Table 2 and 4: in the footnotes of those two tables it is reported that “The significant results (instrument variable regression p-value < 0.05 and Wu-Hausman test p-value ≥0.05) are considered false positives and marked with strikethroughs; but there are no strikethrough lines; at least the first line of Table 2 should be crossed out, following what is indicated in the text at lines 216-219 of paragraph 3.2.1.
Response: We sincerely apologize that the strikethroughs were omitted in the submitted files. We added them where necessary.
Reviewer 2 Report
Ms ID: ijerph-1555928
Title: “Prenatal exposure to traffic-related air pollution and the DNA methylation in cord-blood cells: MOCEH Study”
Authors: Jaehyun Park, Woo Jin Kim, Jeeyoung Kim, Chan-Yeong Jeong, Hyesook Park, Yun-Chul Hong, Mina Ha, Yangho Kim, Sungho Won, Eunhee Ha
Overview:
The manuscript describes the authors’ investigation of the effects of prenatal exposure to air pollution on DNA-methylation in cord blood, more specifically they assessed associations between PM10 and NO2 exposures and CpG methylation. This study is of importance, as it uses cord blood samples from a Korean cohort to investigate the impact of air pollution on infants, a population for which data is so far lacking. It furthermore adds valuable information to the body of evidence on epigenetic effects of NO2 and PM10 exposure. The manuscript is overall well written and organised and the contents fit within the remit of the journal.
Whilst the manuscript is of relevance and provides important information, there are some comments and questions which are addressed below.
Major comments and questions:
The title and abstract are appropriate and reflect the contents of the manuscript. The introduction is short and concise, but contains the relevant information and background for the study.
The materials and methods describe the study population and is strong on the data processing and statistical side. However, there is some information missing on sampling procedures and biochemistry. Could the authors please provide some brief information how the cord blood was processed for epigenetic analysis and also in terms of leukocyte analysis, which was assessed as adjusting covariate, and for which the estimates are shown in table 1? And when was the maternal urine sample provided for cotinine measurement and were any other parameters assessed? The authors provide information on the quality control of the methylation assay in section 2.4. Would it be possible to provide the outcomes of the quality checks, for instance in the supplement, to show the performance of the assay?
The results section is well structured, and the findings are appropriately presented. Specific comments are listed below. Section 3.4 mentions supplementary materials on phenotypes related to the DMPs. However, the only supplementary information that was provided are tables 1 and 2. Could the authors please clarify and/or provide the additional information? In section 3.5, could the authors please include all significant findings from the table, i.e. include GO:0051495?
The discussion puts the findings in context of other studies and addresses its limitations. With regards to other studies, did the authors also look for studies that investigated other proxies for prenatal exposures such as placenta? And how do the findings compare with methylation studies in adults?
Specific comments are listed below.
Specific comments:
Page 1, line 41: Please capitalise M in “HumanMethylation”.
Page 2, line 53: The abbreviation TRAP is only used here, thus it is not necessarily required.
Page 2, line 93: I suggest to use "determination” or “assessment” of air-pollutant concentration. “Measurement” suggests an analytical measurement method.
Page 3, line 114: There seems to be a full stop missing after “procedure”.
Page 5, line 212: Please write out VIF.
Page 7, lines 261ff: Could you please revise this section. It states no significant p-values for the individual pollutants but more significant correlations with NO2 than PM10. I understand this to mean that he p-values for NO2 are lower, but the section is not clear. Also, a bit more information in the text on the findings summarised in table 4 would be welcome, as these seem to be the largest set of DMPs.
Page 10, line 332: Please remove the full stop after “island”.
Table 1: Please check the formatting and footnotes. The Pilcrow and the Star are not shown in the footnotes.
Tables 2 to 5: please check that the appropriate values are in bold or strikethrough. Particularly for the latter there seems to be a discrepancy between the results and the table. There is also some duplicated information in the footnotes, e.g. that significant values are in bold. Please check and clear up.
General comments on the formatting: There are some issues with missing spaces in the text, e.g. before citations. This may be due to the PDF conversion, but I wanted to point it out.
The submission had 2 supplements, one called “non-published”. However, these files appear to be identical. Please check if there is some information missing or if it is just a duplication of the supplement.
Author Response
Overview:
The manuscript describes the authors’ investigation of the effects of prenatal exposure to air pollution on DNA-methylation in cord blood, more specifically they assessed associations between PM10 and NO2 exposures and CpG methylation. This study is of importance, as it uses cord blood samples from a Korean cohort to investigate the impact of air pollution on infants, a population for which data is so far lacking. It furthermore adds valuable information to the body of evidence on epigenetic effects of NO2 and PM10 exposure. The manuscript is overall well written and organized and the contents fit within the remit of the journal. Whilst the manuscript is of relevance and provides important information, there are some comments and questions which are addressed below.
Major comments and questions:
The title and abstract are appropriate and reflect the contents of the manuscript. The introduction is short and concise, but contains the relevant information and background for the study.
Response: according to reviewer’s comments, we have described DNA extraction process in 2.1 section and assessment of DNA methylation levels in 2.4 section. In terms of leucocyte analysis, we added text in the 2.3. (Line 155-156), as given below:
“2.1: “383 cord blood samples corresponding to 195 baby boys and 188 baby girls were selected to be undergone DNA methylation profiling and finally included in the present study.”
2.3:” Pre-processing of the information on the cord blood samples, leukocyte-composition estimation, and quality-control steps were performed with the R packages ewastools (Heiss and Just, 2018; Heiss and Just, 2019) and minfi (Aryee et al, 2014) as described previously (Park et al, 2021). Briefly, leukocyte compositions were estimated with Houseman’s method, with the reference panel by Salas et al. with IDOL-optimized CpGs. Quality control steps were performed by (1) BeadArray Controls metrics by Illumina, (2) sex discordance by the probe intensities, (3) outliers or duplicates by SNP probes, (4) outliers by principal components (PCs), and (5) outliers by leukocyte compositions. The quality control results are shown in the Supplementary Table 1.”
2.4: “The methylation levels of the cord blood samples were determined using Illumina HumanMethylationEPIC BeadChip with the manufacturer’s instructions. Genotyping of the samples was performed as previously described (Park et al. 2020). Briefly, cord-blood samples were genotyped on Asian Precision Medicine Research Array (APMRA; Affymetrix, CA, USA). Quality control was conducted using plink v1.90b3.44(Chang et al. 2015)and WISARD 1.3.3, with the criteria of X chromosome inbreeding coefficients, missing rates, identity-by-state (IBS) matrix, multidimensional scaling (MDS) plots, Hardy-Weinberg equilibrium (HWE), and minor-allele frequencies. The rest of the missing values were filled with SHAPEIT v2.r837(Delaneau et al. 2011), and the data were then imputed to the whole-genome scale by using IMPUTE 2.3.2 with the phase-3 reference data from the 1000 Genome Project(Howie et al. 2012; Howie et al. 2011; Howie et al. 2009). The same quality-control procedures were conducted once again after the imputation step. The results of the quality control step is described in the Supplementary Table 2.”
And when was the maternal urine sample provided for cotinine measurement and were any other parameters assessed?
Response: Prenatal examination and collection of biological samples including cotinine measurement was performed during the first and third trimesters, as described in section 2.1. (Line 92-93) , as follows: “Urinary cotinine and creatinine in the pregnant women was measured during their prenatal visits (early and late pregnancy). ”
The authors provide information on the quality control of the methylation assay in section 2.4. Would it be possible to provide the outcomes of the quality checks, for instance in the supplement, to show the performance of the assay?
Response: We have added the results of the quality control steps in the supplementary document as table 2.
The results section is well structured, and the findings are appropriately presented. Specific comments are listed below. Section 3.4 mentions supplementary materials on phenotypes related to the DMPs. However, the only supplementary information that was provided are tables 1 and 2. Could the authors please clarify and/or provide the additional information?
Response: In this section, the term “results” is confusing, so we replaced it to data.
In section 3.5, could the authors please include all significant findings from the table, i.e. include GO:0051495?
Response: We removed this section as reviewer 3 suggested.
The discussion puts the findings in context of other studies and addresses its limitations. With regards to other studies, did the authors also look for studies that investigated other proxies for prenatal exposures such as placenta? And how do the findings compare with methylation studies in adults?
Response: We added text on prenatal exposures and placental DNA methylation, as follows:
“Furthermore, placental DNA methylation is another organ reflecting prenatal environmental exposure and potentially related to health outcomes, which can be compared to cord blood results. (Ghazi et al. 2021)” (Line 362-364)
Specific comments are listed below.
Specific comments:
Page 1, line 41: Please capitalize M in “Human Methylation”.
Response: We capitalized M in “Human Methylation “.
Page 2, line 53: The abbreviation TRAP is only used here, thus it is not necessarily required.
Response: We have removed the abbreviation.
Page 2, line 93: I suggest to use "determination” or “assessment” of air-pollutant concentration. “Measurement” suggests an analytical measurement method.
Response: We changed measurement to assessment.
Page 3, line 114: There seems to be a full stop missing after “procedure”.
Response: We added full stop after “procedure”.
Page 5, line 212: Please write out VIF.
Response: We wrote the full form of VIF.
Page 7, lines 261ff: Could you please revise this section. It states no significant p-values for the individual pollutants but more significant correlations with NO2 than PM10. I understand this to mean that the p-values for NO2 are lower, but the section is not clear. Also, a bit more information in the text on the findings summarized in table 4 would be welcome, as these seem to be the largest set of DMPs.
Response: We revised the sentence to be more precise (NO2 had lower p-values than PM10). Also, we added some explanation of the findings in table 6 (former table 4) in the main text, as follows”
“We observed more significant correlations with NO2 than PM10 concentrations in individual analyses. cg06517429 (p-value 3.47 × 10-8), one of the main findings from the analyses of whole pregnancy, was also found to be significantly correlated to air pollutants in the second trimester. The most significant sites include cg04129282 (p-value 4.37 × 10-9) on chromosome 15, annotated with WDR93 and PEX11A, and cg06772824 (p-value 4.84 × 10-9) on chromosome 2, annotated with FAM176A.”
Page 10, line 332: Please remove the full stop after “island”.
Response: We removed full stop after “island”.
Table 1: Please check the formatting and footnotes. The Pilcrow and the Star are not shown in the footnotes.
Response: According to reviewer’s comment, we separated table 1 into 2 tables and removed the Pilcrow and the Star.
Tables 2 to 5: please check that the appropriate values are in bold or strikethrough. Particularly for the latter there seems to be a discrepancy between the results and the table. There is also some duplicated information in the footnotes, e.g. that significant values are in bold. Please check and clear up.
Response: We arranged the sentences and moved the footnotes to the table explanation. Also, we sincerely apologize that the strikethroughs were omitted in the submitted version. We have added the strikethroughs in the table.
General comments on the formatting: There are some issues with missing spaces in the text, e.g. before citations. This may be due to the PDF conversion, but I wanted to point it out.
The submission had 2 supplements, one called “non-published”. However, these files appear to be identical. Please check if there is some information missing or if it is just a duplication of the supplement.
Reviewer 3 Report
The manuscript by Park et al. entitled "Prenatal exposure to traffic-related air pollution and the DNA methylation in cord-blood cells: MOCEH Study" revisits the potential association between air pollution exposure during pregnancy and changes in cord blood DNA methylation in the newborn. This time the authors use the MOCEH Korean birth cohort (n=384) and used single and multipollutant approaches to understand potential associations. The manuscript explores this important association to a group of pollutants that are ubiquitous and with previous evidence from large birth cohort meta-analyses (Gruzieva et al, 2017, 2019).
Comments:
Lines 41, 91, 197: Could the authors confirm the number of samples? Is that a typo on the abstract? Given that only 358 were finally used, that number will be more adequate for the abstract. Please correct or add information as needed.
Lines 94-112: This is not my main area of research, so this may be a naive question, but for me it is unclear. How did the LUR convert the total air pollution exposure to traffic air pollution exposure? I see the model adjusted for traffic load and intensity in the buffer zone. Is that translated into traffic-related exposure? if so, how is that compared to the total air pollution exposure for the participants? could you please clarify for me and the readers?
Lines 133-142: Please add the references for missForest, ewastools, minfi, and the specific reference used for cell deconvolution (e.g., Bakulski, Gervin, Gervin/Salas, etc.). Specify the preprocessing procedure including normalization, filtering, limits of detection, and imputation as needed.
Lines 164-165, lines 212-213: What steps were performed by the authors to control for potential multicollinearity in your multipollutant models? Please explain and add information as needed. What exactly were you trying to control via Bonferroni adjustment? Usually, high collinearity will decrease the power and may generate wrong sign estimates? Was this controlled or adjusted in your adjusted results?
Lines 187-190: Please add the references of the ewascatalog and missMethyl to the manuscript.
Table 1: The amount of natural killer cells seems rather low in your cohort (~3%). That is unusual and for such a large cohort it seems biologically implausible (NKs could in some cases be as high as 25%). Could you please check which method was used for cell deconvolution? If you are using any of the older methods (Bakulski), I would recommend that you switch to a cleaner more contemporary method of cell deconvolution (see Gervin, Salas, et al. 2019 for details about this potential problem).
Table 2: What do you mean with "marked with strikethrough" in your caption? Is this a typo?
Table 2, 3, 4, and 5: when you refer to the p-value (all pollutants) where is this P-value coming from? Is this the F-test for the regression? Similarly, when you refer to the p-value for NO2 and PM10, are these the p-values of the specific coefficients in the multivariable regression? In the text, you refer to these as p-values of correlation. Could you please clarify for the reviewer and the readers?
Lines 307-316, Table 6, lines 343-350: I respectfully disagree with the authors here. If you read the table, there is only one differentially methylated site in the four GO pathways, so for instance 1 out of five genes will be statistically significant, but it won't have a biological meaning for that pathway. In my opinion, these are all false positives, I would encourage the authors to remove these results from the manuscript as they are misleading.
Lines 334-335: I have trouble following the logic here. Why are you mentioning nitrites in this context? Is there any pathway converting nitric oxide into nitrites in this context? I know that biologically this is possible through ceruloplasmin, but I thought most of the nitroso oxides will be highly acidic not necessarily going through that specific pathway. Could you help me and the readers to connect the dots here?
Lines 354-366: This is outstanding in your study, but it may need some additional explanation about the relation between PM10 and NO2 exposure in your cohort. How correlated were these measures?
Author Response
The manuscript by Park et al. entitled "Prenatal exposure to traffic-related air pollution and the DNA methylation in cord-blood cells: MOCEH Study" revisits the potential association between air pollution exposure during pregnancy and changes in cord blood DNA methylation in the newborn. This time the authors use the MOCEH Korean birth cohort (n=384) and used single and multipollutant approaches to understand potential associations. The manuscript explores this important association to a group of pollutants that are ubiquitous and with previous evidence from large birth cohort meta-analyses (Gruzieva et al, 2017, 2019).
Comments:
Lines 41, 91, 197: Could the authors confirm the number of samples? Is that a typo on the abstract? Given that only 358 were finally used, that number will be more adequate for the abstract. Please correct or add information as needed.
Response: Thank you for reviewer’s comment. The number 384 in the abstract mean, the number of samples “before” the quality control i.e. all of the samples included in the chip-analysis. We removed the term “with 384 samples” to prevent the confusion.
Lines 94-112: This is not my main area of research, so this may be a naive question, but for me it is unclear. How did the LUR convert the total air pollution exposure to traffic air pollution exposure? I see the model adjusted for traffic load and intensity in the buffer zone. Is that translated into traffic-related exposure? if so, how is that compared to the total air pollution exposure for the participants? could you please clarify for me and the readers?
Response: Thank you for reviewer’s comments and the explanation of LUR was added in the methods section (Line 118-123) and equation of modeling is given below:
“LUR analyzes traffic-related air pollution exposure values through a multiple linear regression modeling method between predictors that can best explain traffic-related air pollution concentrations among variables such as total road length and traffic intensity on roads. In the modeling process, we analyzed how the air pollution concentration value was predicted through the LOOCV method and RMSE confirmation (Sarah et al. 2007).”
NO2 concentration = β0 + β1x length of all the roads + β2x traffic intensity + β3x heavy-duty traffic load + β4xurban green area + β5x urban green area + β5xspatial trend variable
PM10 concentration = β0 + β1x length of all the roads + β2x heavy-duty traffic load + β3xspatial trend variable
- Reference: Sarah et al. Application of Land Use Regression to Estimate Long-Term Concentrations of Traffic-Related Nitrogen Oxides and Fine Particulate Matter, Environ. Sci. Technol. 2007, 41, 2422-2428
Lines 133-142: Please add the references for missForest, ewastools, minfi, and the specific reference used for cell deconvolution (e.g., Bakulski, Gervin, Gervin/Salas, etc.). Specify the preprocessing procedure including normalization, filtering, limits of detection, and imputation as needed.
Response: Since the preprocessing procedures are almost identical to our previous paper (Park et al. 2021; https://doi.org/10.1016/j.envres.2021.110767), we omitted the details. We added the explanations of the processing, as follows:
“Briefly, leukocyte compositions were estimated with Houseman’s method, with reference panel by Salas et al. with IDOL-optimized CpGs. Quality control steps were performed by (1) BeadArray Controls metrics by Illumina, (2) sex discordance by the probe intensities, (3) outliers or duplicates by SNP probes, (4) outliers by principal components (PCs), and (5) outliers by leukocyte compositions.”(Line 155-160)
Lines 164-165, lines 212-213: What steps were performed by the authors to control for potential multicollinearity in your multipollutant models? Please explain and add information as needed. What exactly were you trying to control via Bonferroni adjustment? Usually, high collinearity will decrease the power and may generate wrong sign estimates? Was this controlled or adjusted in your adjusted results?
Response: Thank you for very important comment, we checked the correlations among covariates as follows:
|
no2_1T |
no2_2T |
no2_3T |
no2_preg |
pm10_1T |
pm10_2T |
pm10_3T |
pm10_preg |
|
|
no2_1T |
1.000 |
0.797 |
0.679 |
0.894 |
0.542 |
0.398 |
0.046 |
0.581 |
|
no2_2T |
1.000 |
0.828 |
0.958 |
0.067 |
0.510 |
0.414 |
0.554 |
|
|
no2_3T |
1.000 |
0.905 |
0.035 |
0.082 |
0.575 |
0.367 |
||
|
no2_preg |
1.000 |
0.224 |
0.372 |
0.377 |
0.548 |
|||
|
pm10_1T |
1.000 |
0.227 |
-0.334 |
0.561 |
||||
|
pm10_2T |
1.000 |
0.115 |
0.782 |
|||||
|
pm10_3T |
1.000 |
0.387 |
||||||
|
pm10_preg |
1.000 |
As can be seen from this table, multipollutant are not highly correlated our data. Variance inflation factors ranged from 1.40 to 1.51, much less than 10. Therefore, we concluded that multicollinearity is not substantially large and does not affect the statistical power for our analyses. Bonferroni adjustment was applied for multiple testing problems generated by the large number of CpG markers. Bonferroni adjustment is the most conservative method and the significant result at the 0.05 indicates strong significance for our result. We added the following text in manuscript.
“The correlations between NO2 and PM10 measured by Pearson’s and Spearman’s correlation coefficients in each trimester of pregnancy were around 0.5, and the variance inflation factors for those variables from the multipollutant linear models ranged from 1.40 to 1.51. Therefore, we judged that the multicollinearity would not interrupt the performance of the analyses. As many of the analyses showed deflated p-values, we adjusted the p-values via Bonferroni’s method, which is the most conservative method.” (Line 236-241)
Lines 187-190: Please add the references of the ewascatalog and missMethyl to the manuscript.
Response: We have added references of the ewascatalog and missMethyl .
Table 1: The amount of natural killer cells seems rather low in your cohort (~3%). That is unusual and for such a large cohort it seems biologically implausible (NKs could in some cases be as high as 25%). Could you please check which method was used for cell deconvolution? If you are using any of the older methods (Bakulski), I would recommend that you switch to a cleaner more contemporary method of cell deconvolution (see Gervin, Salas, et al. 2019 for details about this potential problem).
Response: The deconvolution method was described in our previous paper (Park et al. 2021; https://doi.org/10.1016/j.envres.2021.110767). Briefly, we used Houseman’s method, with reference panel by Salas (2018) with IDOL-optimized CpGs. Usage of IDOL standardized library selection is described in Gervin, Salas, et al.
We may look into this problem in more detail by using other deconvolution methods/libraries and compare the results. Unfortunately, due to the shortage of time, we could not able to perform additional analyses with other deconvolution methods. We have added this in the discussion section.
“We found that the proportion of natural killer cells in our samples were relatively low (3.31%; Table 2). This may be due to the selection of the leukocyte deconvolution methods, and it may differ by the method or the reference panel. Unfortunately, we could not test other deconvolution methods and compare the results due to the shortage of time. Further studies may investigate the effect of the methods to the overall results.” (Line 358-364)
Table 2: What do you mean with "marked with strikethrough" in your caption? Is this a typo?
Response: The strikeouts were omitted in the submitted document. We sincerely apologize for that, and we added them in the table.
Table 2, 3, 4, and 5: when you refer to the p-value (all pollutants) where is this P-value coming from? Is this the F-test for the regression? Similarly, when you refer to the p-value for NO2 and PM10, are these the p-values of the specific coefficients in the multivariable regression? In the text, you refer to these as p-values of correlation. Could you please clarify for the reviewer and the readers?
Response: Thank you for the reviewer’s comment The. meaning of the p-values is as you pointed out The p-values for ‘all pollutants’ are from F-test for both of the pollutants, and the p-values for NO2 and PM10 are p-values for regression coefficients from multivariable analysis. We added the explanation in the table description to help interpretation, as follows:
“P-values for “all pollutants” indicate the p-values from F-tests for both of the pollutants, and P-values for each pollutant mean the p-values of the corresponding regression coefficients from the multipollutant model.”
Lines 307-316, Table 6, lines 343-350: I respectfully disagree with the authors here. If you read the table, there is only one differentially methylated site in the four GO pathways, so for instance 1 out of five genes will be statistically significant, but it won't have a biological meaning for that pathway. In my opinion, these are all false positives, I would encourage the authors to remove these results from the manuscript as they are misleading.
Response: According to the reviewer’s comment, we have removed the results from manuscript.
Lines 334-335: I have trouble following the logic here. Why are you mentioning nitrites in this context? Is there any pathway converting nitric oxide into nitrites in this context? I know that biologically this is possible through ceruloplasmin, but I thought most of the nitroso oxides will be highly acidic not necessarily going through that specific pathway. Could you help me and the readers to connect the dots here?
Response: Thank you for reviewer’s comment. We agree that mentioning nitrites was confusing and unnecessary, and we deleted the corresponding sentence.
Lines 354-366: This is outstanding in your study, but it may need some additional explanation about the relation between PM10 and NO2 exposure in your cohort. How correlated were these measures?
Response: The pollutants showed their correlations (Pearson’s & Spearman’s coefficient) about 0.5, and they showed weak multicollinearity i.e. variance inflation factor from 1.41 to 1.50. In the main text, we added the following paragraph.
“The correlations between NO2 and PM10 measured by Pearson’s and Spearman’s correlation coefficients in each period were around 0.5, and the variance inflation factors for those variables from the multipollutant linear models ranged from 1.40 to 1.51. Therefore, we judged that the multicollinearity would not interrupt the performance of the analyses.” (Line 236-239)
Round 2
Reviewer 3 Report
The manuscript by Park et al. entitled "Prenatal exposure to traffic-related air pollution and the DNA methylation in cord-blood cells: MOCEH Study" revisits the potential association between air pollution exposure during pregnancy and changes in cord blood DNA methylation in the newborn. This time the authors use the MOCEH Korean birth cohort (n=384) and used single and multipollutant approaches to understand potential associations. The manuscript explores this important association to a group of pollutants that are ubiquitous and with previous evidence from large birth cohort meta-analyses (Gruzieva et al, 2017, 2019).
The authors have addressed most of the comments.
My only additional minor comment is that you forgot to add the reference for Gervin, Salas et al. 2019 for the reference used on line 156.
Author Response
Thank you for your valuable comments.
We have added the reference in Line 156, as well as a additional reference for the LUR modeling in Line 120.